# PCSK9 Inhibitors Reduce PCSK9 and Early Atherogenic Biomarkers in Stimulated Human Coronary Artery Endothelial Cells

**DOI:** 10.3390/ijms24065098

**Published:** 2023-03-07

**Authors:** Rahayu Zulkapli, Suhaila Abd Muid, Seok Mui Wang, Hapizah Nawawi

**Affiliations:** 1Institute of Pathology, Laboratory and Forensic Medicine (I-PPerForM), Universiti Teknologi MARA (UiTM), Sungai Buloh Campus, Jalan Hospital, Sungai Buloh 47000, Selangor, Malaysia; 2Faculty of Medicine, Universiti Teknologi MARA (UiTM), Sungai Buloh Campus, Jalan Hospital, Sungai Buloh 47000, Selangor, Malaysia; 3Faculty of Dentistry, Universiti Teknologi MARA (UiTM), Sungai Buloh Campus, Jalan Hospital, Sungai Buloh 47000, Selangor, Malaysia

**Keywords:** PCSK inhibitor, PCSK9, alirocumab, evolocumab, human coronary artery endothelial cell, atherogenesis, atherosclerosis

## Abstract

Despite reports on the efficacy of proprotein convertase subtilisin-Kexin type 9 (PCSK9) inhibitors as a potent lipid-lowering agent in various large-scale clinical trials, the anti-atherogenic properties of PCSK9 inhibitors in reducing PCSK9 and atherogenesis biomarkers via the NF-ĸB and eNOS pathway has yet to be established. This study aimed to investigate the effects of PCSK9 inhibitors on PCSK9, targeted early atherogenesis biomarkers, and monocyte binding in stimulated human coronary artery endothelial cells (HCAEC). HCAEC were stimulated with lipopolysaccharides (LPS) and incubated with evolocumab and alirocumab. The protein and gene expression of PCSK9, interleukin-6 (IL-6), E-selectin, intercellular adhesion molecule 1 (ICAM-1), nuclear factor kappa B (NF-ĸB) p65, and endothelial nitric oxide synthase (eNOS) were measured using ELISA and QuantiGene plex, respectively. The binding of U937 monocytes to endothelial cell capacity was measured by the Rose Bengal method. The anti-atherogenic effects of evolocumab and alirocumab were contributed to by the downregulation of PCSK9, early atherogenesis biomarkers, and the significant inhibition of monocyte adhesion to the endothelial cells via the NF-ĸB and eNOS pathways. These suggest the beyond cholesterol-lowering beneficial effects of PCSK9 inhibitors in impeding atherogenesis during the initial phase of atherosclerotic plaque development, hence their potential role in preventing atherosclerosis-related complications.

## 1. Introduction

Various atherogenesis biomarkers of early atherosclerosis have been proposed to predict cardiovascular events and mechanisms. The early stage of atherosclerosis development involves inflammation, endothelial injury, and endothelial activation before the formation of atherosclerotic plaques [1]. The overexpression of proinflammatory cytokines (interleukin-6; IL-6) and adhesion molecules (intercellular adhesion molecule 1; ICAM-1 and E-selectin) by endothelial cells, mediated via nuclear factor-kappa beta (NF-ĸB) activation, will occur before the adherence of circulating monocytes on endothelial cells [2,3,4]. Moreover, inflammation, endothelial activation, and oxidative stress have been identified as critical events in the initiation and progression of atherosclerosis [5]. 

In July and August 2015, the FDA approved novel lipid-lowering therapy (LLT), namely alirocumab (human IgG1/κ mAb, genetically engineered in Chinese hamster ovary cells) and evolocumab (fully human IgG2 mAb), which aimed to reduce low-density lipoprotein cholesterol (LDL-C) [6,7,8]. These novel therapies are monoclonal antibodies that inactivate the proprotein convertase subtilisin-Kexin type 9 (PCSK9) enzyme that binds to LDLR to initiate the reuptake and catabolism of low-density lipoprotein cholesterol receptor (LDLR) by body cells. Diminished LDLR will reduce LDL-C uptake from the blood into the cells, causing increased LDL-C levels in the blood and leading to an increased risk of coronary artery disease (CAD). 

The PCSK9 inhibitor is a lipid-lowering agent prescribed for severe hypercholesterolemia patients. Alirocumab and evolocumab have shown striking LDL-C reductions of up to 60% and are documented to be relatively safe [6,9,10,11]. PCSK9-bound LDLR will degrade within the cell cytoplasm, thus preventing it from being recycled to the cell surface for LDL-C reuptake [12,13]. In normal conditions, PCSK9 is co-expressed with LDLR by the same transcription factor (sterol-regulatory element-binding protein 2) [14] to modulate the cellular reuptake of circulatory LDL-C so that the LDL-C in plasma is not depleted by the cells. By inhibiting the PCSK9, the LDLR will reuptake the circulatory LDL-C unhindered, significantly reducing the LDL-C level [15,16]. 

There are various reports on the efficacy of PCSK9 inhibitors in LDL-C lowering in a variety of comprehensive large-scale clinical trials such as PROFICIO (Program to Reduce LDL-C and Cardiovascular Outcomes Following Inhibition of PCSK9 in Different Populations) [17,18,19,20,21,22], FOURIER (Further Cardiovascular Outcomes Research With PCSK9 Inhibition in Subjects with Elevated Risk) [23,24,25] and ODYSSEY (Cardiovascular Outcomes After an Acute Coronary Syndrome) [26,27,28]. PCSK9 inhibition attenuates atherosclerosis progression and lowers the risk for acute cardiovascular events [29,30,31]. In addition, preclinical studies also revealed that PCSK9 has pleiotropic effects other than plasma LDL-C regulation and could be a key molecule in the pathophysiology of atherosclerosis [32]. However, the molecular mechanisms through which PCSK9 inhibition might confer atheroprotection beyond low-density lipoprotein (LDL) lowering remain scarce. Furthermore, inflammation-stimulating agents, lipopolysaccharides (LPS), have been shown to increase PCSK9 expression significantly and act as an inflammatory trigger, which causes an increase in the uptake of oxidized LDL into the subendothelial space [33,34,35], mimicking the process of atheroma formation in the blood vessels. 

Therefore, this study aimed to investigate the effects of PCSK9 inhibitors on PCSK9 and early atherogenesis biomarker expression of inflammation, endothelial activation, oxidative stress, and monocyte-endothelial binding in LPS-stimulated HCAEC compared to controls. 

## 2. Results

### 2.1. Cytotoxic Effects of PCSK9 Inhibitors on HCAEC

The incubation of LPS-stimulated HCAEC with PCSK9 inhibitors (evolocumab or alirocumab) at 1, 10, and 100 µg/mL for 24 h exhibited more than 90% cell viability (Figure 1). Similarly, the incubation of LPS-stimulated HCAEC with media (untreated) and LPS alone (negative control) for 24 h also did not show any reduction in cell viability (Figure 1). Therefore, the selected concentrations of PCSK9 inhibitors were regarded as non-toxic to HCAEC, and they were subsequently used on HCAEC in this study.

### 2.2. Effects of PCSK9 Inhibitors on PCSK9 Protein Expression

PCSK9 inhibitors exhibit significant reducing effects on PCSK9 protein suppression on LPS-stimulated HCAEC compared to LPS control (Figure 2a). Both exhibited a reduction in PCSK9 protein expression across all concentrations (1–100 µg/mL) (*p* < 0.05 and *p* < 0.001). Alirocumab and evolocumab work best at the lowest concentration tested (1 µg/mL) in downregulating PCSK9 protein expression (*p* < 0.001, *p* < 0.05). According to AUC analysis, alirocumab (14.0%) is the most potent PCSK9 inhibitor for inhibiting PCSK9 protein expression compared to evolocumab (5.7%) (Table 1, Figure 2a).

### 2.3. Effects of PCSK9 Inhibitors on Endothelial Activation Protein Expression (ICAM-1 and E-Selectin)

There was a significant reduction in endothelial activation protein expression by the co-incubation with a PCSK9 inhibitor at different concentrations. Specifically, evolocumab (10 µg/mL) and alirocumab (100 µg/mL) suppressed ICAM-1 significantly compared to the LPS control (*p* < 0.01) (Figure 2b). Similarly, LPS stimulated-HCAEC treated with alirocumab or evolocumab showed E-selectin reduction at various concentrations, suggesting both inhibitors exhibited dose-independent effects. Evolocumab evidenced suppression only at the highest concentration tested, 100 µg/mL (*p* < 0.01). Alirocumab reduced E-selectin at all concentrations tested except for 10 µg/mL (*p* < 0.01). The most potent PCSK9 inhibitor for E-selectin protein expression inhibition was alirocumab (11.9%), followed by evolocumab (3.7%) (Table 1, Figure 2c).

### 2.4. Effects of PCSK9 Inhibitors on IL-6 Protein Expression

Alirocumab and evolocumab, across all concentrations (1–100 µg/mL), showed no significant reductions in IL-6 protein expression compared to LPS-stimulated HCAEC (Figure 2d). Instead of downregulating IL-6, IL-6 protein expression was upregulated with a PCSK9 inhibitor (*p* < 0.05, *p* < 0.001) compared to the LPS controls. However, the untreated and unstimulated cells show a very minimal IL-6 protein expression compared to the LPS control, which ensures the study’s validity.

### 2.5. Effects of PCSK9 Inhibitor on PCSK9 Gene Expression

The co-incubation of evolocumab at 1 and 10 µg/mL with LPS-stimulated HCAEC significantly downregulated the *PCSK9* mRNA (*p* < 0.05 and *p* < 0.01, respectively). On the contrary, instead of downregulating the level of PCSK9 expression, Alirocumab caused an upregulation of *PCSK9* mRNA across all concentrations compared to LPS controls (Figure 3a).

### 2.6. Effects of PCSK9 Inhibitor on Endothelial Activation Gene Expression (ICAM-1 and E-Selectin)

The co-incubation of LPS and evolocumab down-regulated both endothelial activation gene expression, *ICAM-1* (*p* < 0.01 and *p* < 0.05) and *E-selectin* (*p* < 0.05) mRNA at 10 and 100 µg/mL, respectively. In contrast, alirocumab enhanced the endothelial activation gene expression above LPS controls (*p* < 0.05) (Figure 3b,c). Thus, the AUC analysis only showed the percentage of inhibition of evolocumab for downregulating *ICAM-1* and *E-Selectin* gene expression (57.4% and 126.9%, respectively) (Table 1).

### 2.7. Effects of PCSK9 Inhibitor on IL-6 Gene Expression

All concentrations of both PCSK9 inhibitors did not suppress *IL-6* mRNA expression when compared to LPS controls (Figure 3d), which was consistent with IL-6 protein expression (Figure 2d).

### 2.8. Effects of PCSK9 Inhibitor on NF-ĸB p65 Protein and Gene Expression

Evolocumab and alirocumab showed a reduction trend of NF-κB p65 protein compared to LPS controls, although not a statistically significant one. It can be seen through AUC analysis that evolocumab (21.6%) inhibits the NF-κB p65 protein expression at a higher percentage than alirocumab (16.0%) (Table 1). The negative control showed an opposite trend of NF-κB p65 protein expression (up-regulating) (Figure 4a). The same applies to gene expression; the co-incubation of LPS with evolocumab at 10 μg/mL down-regulated the mRNA level of *NF-κB p65* but was not significant. However, instead of downregulating, alirocumab enhanced *NF-κB p65* gene expression compared to LPS alone at all different concentrations (1–100 µg/mL) (Figure 4b).

### 2.9. Effects of PCSK9 Inhibitor on eNOS Protein and Gene Expressions

Co-incubation of LPS alone reduces eNOS in terms of protein and gene expressions. Meanwhile, the unstimulated/untreated cells showed elevated protein (*p* < 0.001) and gene expression (*p* < 0.001). PCSK9 inhibitors enhanced eNOS protein expressions at different concentrations, where evolocumab at 10 μg/mL (*p* < 0.05) and alirocumab at 1 μg/mL (*p* < 0.05) compared to the LPS control. The potency of both PCSK9 inhibitors in inhibiting eNOS protein is almost similar, where evolocumab was at 18.5% and alirocumab was at 18.6% (Figure 5a, Table 1). In contrast with gene expression, only evolocumab showed an upregulation trend of *eNOS* mRNA at 1–100 µg/mL, although it was not significant. Alirocumab, on the other hand, showed the opposite trend against evolocumab. The most potent PCSK9 inhibitor for upregulating the eNOS biomarker was alirocumab (63.5%) compared to evolocumab (18.6%) (Figure 5b, Table 1).

### 2.10. Effects of PCSK9 Inhibitors on Monocytes and LPS-Stimulated HCAEC Interaction

A monocyte adhesion assay was performed to explore the effects of PCSK9 inhibitors on monocytes and endothelial cell interactions. After co-incubation with LPS alone for 24 h, the adhesion of U937 monocytes to HCAEC was markedly increased. The evolocumab treatment at 100 μg/mL (*p* < 0.001) reduced monocyte adhesion to LPS-stimulated HCAEC compared to LPS controls. A significant reduction was observed at 10 and 100 µg/mL (*p* < 0.001) compared to LPS controls for alirocumab. An AUC analysis showed that alirocumab exhibited 51.54% inhibition of monocyte adhesion, whilst for evolocumab it was 29.15%. The control group, monocytic U937, evidenced minimal adherence to the unstimulated HCAEC (Figure 6, Table 1).

## 3. Discussion

At present, there is an abundance of clinical or in vivo studies on PCSK9 inhibitors. Despite the importance of the in vitro studies, there is a scarcity of research related to PCSK9 inhibitors. To the best of our knowledge, this is the first report to describe the anti-atherogenic effects of PCSK9 inhibitors in HCAEC. HCAEC is the most appropriate cell line to be used in anti-atherogenic studies due to several factors: (1) PCSK9 is expressed by the endothelial cell [36]; (2) Atherosclerosis is a disease related to the arteries; and (3) HCAEC is the cell that lines the innermost layer of the coronary artery blood vessels. 

The greater use of multi-LLT regimens is associated with lower LDL-C levels and better outcomes [37]. Statins, which are functionally known as Hydroxymethylglutaryl-CoA (HMG-CoA) reductase inhibitors, have been used as first-line drugs to treat hypercholesterolemia that primarily decreases LDL-C and triglyceride (TG) levels [38,39]. However, statins also increase PCSK9 activity. While statins efficiently lower cholesterol levels, their efficacy diminishes with a rise in PCSK9 activity. Thus, it is paramount to study the mechanism of other LLTs that can reduce LDL-C and PCSK9, such as evolocumab and alirocumab.

The MTS assay demonstrated that PCSK9 inhibitors (evolocumab and alirocumab) up to 100 µg/mL did not exhibit any toxic effects on the viability of HCAEC. PCSK9 inhibitors were considered safe, as the viability was more than 90%. Whilst studies to validate and compare the safety of PCSK9 are lacking, Safaeian et al. (2019, 2020) [40,41] did report on the non-cytotoxic effects of evolocumab on human umbilical vein endothelial cells (HUVEC) at 0.5–100 µg/mL using an MTT (3-[4,5-dimethylthiazol-2-yl]-2,5 diphenyl tetrazolium bromide) assay. Despite the consensus on the cytotoxic effects of PCSK inhibitors, the comparison of the cell viability may be incomparable due to the different cell lines used. 

In general, LPS-stimulated HCAEC, treated with evolocumab and alirocumab, exhibited a reduction of PCSK9, E-selectin, ICAM-1, and NF-ĸB biomarkers at different treatment concentrations. In contrast, for eNOS, the biomarker expression was marginally upregulated but not significant. For the monocyte binding to endothelial, alirocumab showed a higher percentage of inhibition than evolocumab. Therefore, this study suggested that the mechanism that leads to the improvement of endothelial function may be due to anti-PCSK9 and anti-endothelial activation that was mediated via the NF-κB p65 and eNOS pathways. This is in consensus with the study by Di Minno et al., which observed an improvement in endothelial function after evolocumab therapy in patients with familial hypercholesterolemia (FH) [42]. This is also supported by Maulucci et al., who had similar findings in subjects with increased cardiovascular risk [43].

Excess dietary fat consumption increases hepatic PCSK9 expression [44]. PCSK9 could accelerate atherosclerosis through mechanisms beyond the degradation of the hepatic LDLR. Several clinical studies suggested that PCSK9 is involved in atherosclerotic inflammation [45,46]. PCSK9 is ubiquitously expressed in many tissues and cell types [47]. Feingold et al., (2008) [33] reported that LPS stimulation (5 mg/kg body weight, intraperitoneally) in an animal model (Female C57BL/6 mice) increased PCSK9 mRNA levels by 2.5-fold (4 h) and 12.5-fold (38 h). In an in vitro model, Ding et al., (2015) [22] did not specifically report the increase in PCSK9 expression. Still, they reported the dose-dependent relationship between LPS (0.001–1 µg/mL) stimulation and PCSK9 expression on aortic endothelial cells. PCSK9 expression was more significant in smooth muscle cells (SMC) than in endothelial cells (EC) upon LPS stimulation [34,48]. PCSK9 expression in HCAEC is yet to be reported on. Our in vitro study showed the capability of LPS in upregulating the PCSK9 expression compared to the normal control (unstimulated) in HCAEC. The PCSK9 protein and gene expression increased 1.2-fold with LPS stimulation (24 h).

The LPS-stimulated HCAEC were co-incubated with PCSK9 inhibitors to observe the efficacy of PCSK9 reduction using HCAEC in an in vitro model. Evolocumab showed a promising significant downregulation effect on the PCSK9 protein (decreased 1.2-fold) and gene expression (decreased 1.5-fold) compared to the LPS control. For alirocumab, it reduced the PCSK9 protein downregulation (decreased 1.2-fold), but the gene was upregulated. This might be due to the increase of hepatocyte nuclear factor 1α (*HNF1α*) gene transcription, which is critical in regulating *PCSK9* gene transcription [49]. HNF1α promotes PCSK9 transcription by binding with the HNF1 motif, which is located upstream of sterol regulatory element 1 (SRE1) in the PCSK9 promoter [49]. Yang et al. [50] reported the effects of chitosan oligosaccharides, which downregulated the PCSK9 gene, but upregulated HNF1α and the sterol regulatory element-binding proteins (SREBP). However, none of the studies on alirocumab reported on HNF1α. Based on our findings, we postulated that the HNF1α and SREBP might not be the primary PCSK9 regulators for alirocumab.

Endothelial activation has been established as one of the critical events in atherosclerosis initiation and progression [5]. In unstimulated endothelial cells, E-selectin is undetectable [51], unlike ICAM-1, which is present in healthy arteries [52]. Upon stimulation with LPS, *ICAM-1* gene expression did not change at 0.5 h but increased two- to three-fold at 12 h [53]. The E-selectin mRNA and protein increased in the lymphatic endothelial cells with LPS at more than two-fold levels compared with human umbilical vein endothelial cells (HUVEC) [53]. This is in consensus with our findings that showed that evolocumab significantly downregulated the ICAM-1 protein at 10 µg/mL and E-selectin at 100 µg/mL (*p* < 0.01). The gene expression for *ICAM-1* and *E-selectin* was significantly downregulated at the same concentration, 10–100 µg/mL. For alirocumab, the ICAM-1 and E-selectin protein was reduced. However, *ICAM-1* and *E-selectin* gene expression did not reflect the protein expression. ICAM-1 and E-selectin protein expressions were upregulated compared to LPS controls. This discrepancy might be attributed to changes in the regulatory mechanism, which involves the rate of translation and protein breakdown. mRNAs initially translated may later be temporarily repressed [54].

The high expression of endothelial cellular adhesion molecules (CAM) such as ICAM-1 and E-selectin has been consistently observed in atherosclerotic plaques, as it plays a vital role in inducing monocyte recruitment into the intima. E-selectin is an adhesion receptor that slows leukocyte rolling, and its expression is restricted to endothelial cells [51]. This unstable binding is further facilitated by ICAM-1, which promotes transmigration by rearranging the endothelium cytoskeleton and weakening the strength of endothelial cell junctions [55]. ICAM-1 is a transmembrane immunoglobulin protein predominantly expressed on endothelial cells and is crucial in leukocyte recruitment and transmigration [56].

The early stages in the development of atherosclerosis involve inflammation and endothelial activation, thus making the role of inflammation in atherosclerotic plaque formation crucial. The endothelial cells’ over-expression of proinflammatory cytokines and adhesion molecules is mediated via the activation of NF-ĸB during these stages [57]. NF-ĸB is involved in a proinflammatory signalling pathway responsible for expressing proinflammatory genes, including cytokines, chemokines, and adhesion molecules. One of the most highly induced NF-ĸB-dependent cytokines is IL-6 [58]. Unlike other biomarkers, the protein and gene expression for inflammatory biomarkers (IL-6) of both PCSK inhibitors showed an increase in LPS controls. The AUC of alirocumab is −119.2% (IL-6 protein) and −1373.1% (*IL-6* gene). For evolocumab, the AUC is −132.7% (IL-6 protein) and −156.7% (*IL-6* gene). The NF-ĸB p65 showed a reduction trend but was not statistically significant for protein and gene expressions. Therefore, in these cases, it can be deduced that PCSK9 inhibitors may not reduce the IL-6 inflammatory biomarkers through the NF-ĸB p65 pathway. Leucker et al. [59] reported that unchanged inflammatory biomarkers, including high-sensitivity, C-reactive protein, IL-6, interferon-gamma, tumour necrosis factor-alpha, and soluble CD163 were unchanged in dyslipidemia patients treated with evolocumab.

eNOS, when bound to its co-factor, tetrahydrobiopterin (BH4), will produce nitric oxide (NO). Before the development of atherosclerotic plaques, the NO, which serves as an endothelial vasodilator, is impaired. A defect in NO production or activity has been proposed as a significant mechanism of endothelial dysfunction and a contributor to atherosclerosis [60]. In addition to NO, eNOS produces a superoxide anion if its function is altered. This phenomenon is referred to as “eNOS uncoupling”, as eNOS is not coupled with its cofactor and has been found to play an essential role in the process of various cardiovascular diseases [61]. When endothelial cells are under oxidative stress, oxidation of BH4 to dihydrobiopterin (BH2) will occur. BH2 will bind to eNOS, eNOS becomes uncoupled, and, as a result, the production of reactive oxygen species (ROS) will increase [62]. LDL may be oxidized by ROS released by vascular cells within the arterial wall [63]. ROS generation followed the same pattern as PCSK9 expression, and it is dose-dependent in EC and SMC upon stimulation [5]. In this study, as expected, the eNOS protein and mRNA (*p* < 0.001 and *p* < 0.01) in the control groups (untreated/unstimulated) were found to be highest compared to LPS alone. The eNOS protein expression was upregulated with the co-incubation of alirocumab and evolocumab at specific concentrations. The *eNOS* mRNA was only upregulated for evolocumab. However, the upregulation is not significant. Further investigation on the coupling of eNOS is warranted.

Monocyte adhesion to the endothelium, followed by monocyte extravasation, is an initial stage of atherogenesis development. It is an important event in vascular inflammation. Circulating monocytes are critically involved in the progression of atherosclerosis upon migration to the tissue and differentiate into macrophages. The adhesion cascade is a strongly regulated process, and monocyte adherence will increase in number during chronic inflammation. This process is predominantly mediated by cellular adhesion molecules, including ICAM-1 and E-selectin, which are expressed by activated endothelial cells in response to several inflammatory stimuli, including LPS. Evolocumab and alirocumab (1–100 µg/mL) can suppress monocyte adhesion to LPS-stimulated HCAEC in a dose-dependent manner compared to controls. Alirocumab showed the highest reduction in monocyte binding to HCAEC at 100 µg/mL compared to LPS alone. In this study, both PCSK9 inhibitors, alirocumab (51.54%) and evolocumab (29.15%), showed a prominent percentage of monocyte inhibition. Thus, this suggested their pivotal role in mediating the LDL-C lowering effects in PCSK9 inhibitors, parallel with the reduction of ICAM-1 and E-selectin biomarkers in these in vitro studies.

Despite debates on the cost-effectiveness of PCSK9 inhibitors since 2015, a definite advantage of PCSK9 inhibitors is their better tolerance compared to the lipid-lowering drugs used to date. They cause significantly fewer muscle symptoms than statins [64,65]. As these drugs become more widely used in clinical practice, accumulating scientific and clinical data indicate that PCSK9 inhibitors have an excellent safety and tolerability profile, with a low occurrence of side effects. In the FOURIER (2017) study, the percentage of reported adverse events following evolocumab treatment was much lower than in earlier trials. More importantly, there was no significant difference in the number of adverse events between the trial and placebo groups [31].

## 4. Materials and Methods

### 4.1. Cell Culture

HCAEC were obtained from Cell Applications Inc. (San Diego, CA, USA) and were developed from a normal human coronary artery of a 20-year-old Caucasian. The cell lines were grown in the MesoEndo cell growth medium (Cell Application Inc., San Diego, CA, USA) and detached using Accutase (Nacalai Tesque, Kyoto, Japan). The cell lines were incubated in a 90% humidified atmosphere containing 5% CO_2_ (Galaxy 170 R, Eppendorf, Hamburg, Germany). Cells were grown in a 25 or 75-cm^2^ flask (BD Falcon, London, UK). The cells were passaged when they reached more than 80% confluency. All of the experiments conducted in this research used HCAEC from passages five to nine.

### 4.2. Cell Viability

HCAEC cell viability was determined following the MTS assay (3-(4,5-dimethylthiazol-2-yl)-5-(3-carboxymethoxyphenyl)-2-(4sulfophenyl)-2Htetrazolium) that used the CellTiter 96^®^ AQueous One Solution Cell Proliferation Assay kit (Promega, Madison, WI, USA). The HCAEC were seeded into a 96 wells culture plate (1 × 10^4^ cells/well) and incubated overnight at 37 °C for 24 h in humidified 5% CO_2_ before being treated with different concentrations of alirocumab and evolocumab (1, 10, and 100 μg/mL). Next, 10 µL of the MTS reagent (5 mg/mL MTS) was added to each well and incubated for another 4 h at 37 °C. The absorbance was measured at 490 nm using a microplate reader (Agilent, Santa Clara, CA, USA). The experiment was conducted in three biological replicates. The viability of the cells was measured by comparing the treated wells with the control wells and calculated using the following formula:$$\mathrm{Cell}\;\mathrm{viability}\;\left(\%\right)=\frac{\mathrm{Sample}\;\mathrm{absorbance}-\mathrm{Blank}\;\mathrm{absorbance}}{\mathrm{Control}\;\mathrm{absorbance}-\mathrm{Blank}\;\mathrm{absorbance}}\times 100\%$$

### 4.3. Treatment of HCAEC

The PCSK9 inhibitors were diluted into various working concentrations (1, 10, and 100 μg/mL) with culture media. HCAEC were treated with 1, 10, and 100 μg/mL of PCSK9 inhibitors together with 1 μg/mL of LPS, followed by 24 h of incubation at 37 °C in 5% CO_2_. LPS was used to imitate the release of the inflammatory mediator that contributes to the increased release of PCSK9 and the pathological process of atherogenesis.

### 4.4. Samples Collection for Protein Measurement

#### 4.4.1. Cell Culture Supernatant Collection

LPS-stimulated HCAEC was co-incubated with alirocumab and evolocumab at concentrations of 1, 10, and 100 μg/mL for 24 h. The supernatant was collected and centrifuged for 20 min at 1000× *g* at 2–8 °C. The pellet formed at the bottom of the microcentrifuge tube was discarded. The supernatant was used to carry out the assay was kept at −80 °C before use.

#### 4.4.2. Nuclear Lysates Collection (Nuclear Extraction) for NF-ĸB Assay

The nuclear extraction was performed following the manufacturer’s instructions using the CHEMICON^®^ Nuclear Extraction kit (CHEMICON^®^ International, Temecula, CA, USA). The treated cells were washed with 1× PBS solution, followed by the addition of Accutase to detach the cell, and the volume of cells in the cell pellet was counted and estimated using the cell counter. Five cell pellet volumes of ice-cold 1× cytoplasmic lysis buffer containing 0.5 mM DTT and 1/1000 dilution or inhibitor cocktail were added. The cell pellet was resuspended by gently inverting the tube. The cell suspension was incubated on ice for 15 min and centrifuged at 250× *g* for 5 min at 4 °C. The supernatant was discarded, and the cell pellet was resuspended in two volumes of ice-cold 1× cytoplasmic lysis buffer. A syringe with a small gauge needle (27 gauge) was used to draw the cell suspension prepared from the sample tube into the syringe, and the contents were then ejected back into the sample tube. The drawing and ejecting were repeated approximately five times. The disrupted cell suspension was centrifuged at 8000× *g* for 20 min at 4 °C. The supernatant was discarded. The remaining pellet contains the nuclear portion of the cell lysate. The nuclear pellet was resuspended in 2/3 of the original cell pellet volume of ice-cold nuclear extraction buffer containing 0.5 mM DTT and 1/1000 protease inhibitor cocktail. A fresh syringe with a 27-gauge needle was used. A rotator or orbital shaker (low speed) was used to gently agitate the nuclear suspension at 4 °C for 30–60 min. The nuclear suspension was centrifuged at 16,000× *g* for 5 min at 4 °C. The supernatant containing the nuclear lysates was transferred to a fresh tube to carry out the assay.

#### 4.4.3. Cell Lysates Collection for eNOS Assay

The cell lysates were collected following the protocol from Proteintech^®^ on cell and tissue lysate preparation (Proteintech^®^, Rosemont, IL, USA). The treated cells were pelleted by centrifugation for 5 min at 1000× *g* (approximately 2000 rpm) at 4 °C. The cells were washed three times with ice-cold 1× PBS, then a chilled RIPA buffer with a protease inhibitor was added. In general, 100 μL RIPA buffer was added for approximately every 106 cells in the pellet (count cells before centrifugation). The pellet was vortexed occasionally until it homogenized with the buffer and was kept on ice for 30 min. The sample was sonicated to break the cells up and to shear the cell’s DNA. The sonication time was adjusted to 1 min at a power of about 180 watts (in rounds of 10 s sonication/10 s rest for each cycle). The sample was kept on ice during the sonication.

### 4.5. Measurement of Protein Expression

#### 4.5.1. Quantitation of PCSK9, IL-6, ICAM-1, and E-Selectin in the Supernatant

The supernatant’s concentrations of IL-6, soluble ICAM-1, soluble E-selectin, and PCSK9 were performed with a commercially available standard ELISA kit (Elabscience, Houston, TX, USA), according to the manufacturer’s instructions. The absorbance was measured at 450 nm using a microplate reader (Agilent, Santa Clara, CA, USA).

#### 4.5.2. Measurement of NF-κB p65 Protein in Nuclear Lysates Protein

The nuclear lysates collected were quantified for NF-ĸB using the Elabscience^®^ Human NF-ĸB p65 ELISA kit (Elabscience, Affymetrix, Houston, TX, USA). All procedures were performed according to the manufacturer’s instructions. The absorbance of the samples was measured at 450 nm using a microplate reader (Agilent, Santa Clara, CA, USA).

#### 4.5.3. Quantitation of eNOS Protein in Cell Lysates

The eNOS protein concentration in cell lysates was quantitated using the Elabscience^®^ Human eNOS ELISA kit (Elabscience, Houston, TX, USA). All procedures were conducted following the manufacturer’s instructions. The absorbance was read using a microplate reader at 450 nm with a reference wavelength set at 570 nm (Agilent, Santa Clara, CA, USA).

### 4.6. Gene Expression

#### 4.6.1. RNA Extraction

The total RNA was extracted from the treated cell pellets using the RNA extraction kit from Macherey-Nagel (Duren, Germany). RNA purity and concentration were determined by nanodrop. Buffer RA1 (350 µL) was added to the cell pellet and vortexed vigorously. The mixture was placed in the NucleoSpin^®^ filter (violet ring) on a 2 mL collection tube and centrifuged for 1 min at 11,000× *g*. The NucleoSpin^®^ filter (violet ring) was discarded, and 350 µL ethanol (70%) was added to the homogenized lysates and mixed by pipetting up and down (five times). The lysates were pipetted up two to three times and then transferred to a NucleoSpin^®^ RNA Column (light blue ring) in a collection tube. The column was centrifuged for 30 s at 11,000× *g*, and the collection tube was discarded. The column was placed in a new collection tube. A three hundred fifty µL membrane desalting buffer (MDB) was added and centrifuged at 11,000 for 1 min to dry the membrane. The Dnase reaction mixture (95 µL) was applied directly onto the center of the silica membrane of the column and incubated at room temperature for 15 min. Buffer RAW2 (200 µL) was added to a NucleoSpin^®^ RNA column and centrifuged at 11,000 for 30 s. The column was placed in a new collection tube. Buffer RA3 (600 µL) was added to a NucleoSpin^®^ RNA column and centrifuged at 11,000 rpm for 30 s. The flowthrough was discarded, and the column was placed back into a 2 m collection tube. Buffer RA3 (250 µL) was added to a NucleoSpin^®^ RNA column and centrifuged at 11,000 rpm for 2 min to dry the membrane completely. The column was placed into a nuclease-free collection tube. The RNA was eluted in 60 µL Rnase-free H_2_O and centrifuged at 11,000 rpm for 1 min.

#### 4.6.2. Quantitation of PCSK9, Inflammation, Endothelial Activation, NF-κB, and eNOS Genes

A QuantiGene Plex 96-well assay (Thermofisher Scientific, Waltham, MA, USA) measured gene expression according to the manufacturer’s protocol. In triplicates, RNA was transferred to the assay hybridization plate, and contained a working bead mix and probe sets. Hybridization was performed for 20 h at 54 °C ± 1 °C, shaking at 600 rpm. Next, the mixtures were transferred to a 96-well magnetic separation plate. The beads were hybridized with a preamplifier probe, an amplifier probe, a label probe, and Streptavidin conjugated R-Phycoerythrin (SAPE). SAPE fluorescence was measured with the Luminex FlexMap three-dimensional instrument (Luminex Corporation, Austin, TX, USA) to indicate the volume of mRNA transcripts captured by the beads. Fold-changes will be taken as the relative ratios between the normalized reference values of all treatment groups and the untreated group’s values. The target-specific RNA molecules of LPS-stimulated HCAEC were PCSK9: NM_174936; IL-6: NM_000600; ICAM-1:NM_000201; E-Selectin: NM_000450; NF-ĸB p65: NM_003998; eNOS: NM_000603.

#### 4.6.3. Monocyte Binding Assay

The binding of U937 monocytes to the endothelial cell capacity was measured by the Rose Bengal method [66]. HCAEC were seeded in 96-well microtiter plates at a seeding density of 1 × 10^5^ cells/mL and incubated overnight in a humidified incubator set at 37 °C and 5% CO_2_. Cells were treated with alirocumab and evolocumab at concentrations of 1, 10, and 100 μg/mL for 24 h with 5 × 10^5^ cells/mL monocytes U937 added afterwards and incubated for 1 h at 37 °C. After three washes to remove non-adherent monocytes, 0.25% of Rose Bengal stain in phosphate-buffered saline (PBS) was added to each well for 10 min at 25 °C. The excess stains were washed away three times with PBS [supplemented with 10% fetal bovine serum (FBS)], and the stain was released from the cells with a solution of ethanol: PBS (1:1 *v/v*) for 1 h at 25 °C. Monocyte-endothelial cell adhesion was calculated from the difference in absorbance at 570 nm between wells that contain monocytes and HCAEC and wells that contain HCAEC only.

#### 4.6.4. Statistical Analysis

Results were reported as mean ± standard deviation (SD). All results were analyzed using IBM SPSS Statistic 26. A one-way analysis of variance (ANOVA) followed by a Bonferroni post hoc analysis was used. Two-sided *p* < 0.05 was considered significant. The percentage (%) of inhibition against LPS controls for each biomarker was obtained from the area under the curve (AUC) analysis using Graph Version 4.3.

## 5. Conclusions

This in vitro study demonstrated that the anti-atherogenic properties of PCSK9 inhibition are mediated by endothelial activation and the capability of PCSK9 inhibitors to suppress the binding of monocytes to endothelial cells. These two factors are essential to atheroma formation. These findings will give researchers and pharmaceutical companies a broader view in developing prospective medications and providing better management plans for hypercholesterolemia patients.

## Figures and Tables

**Figure 1 ijms-24-05098-f001:**
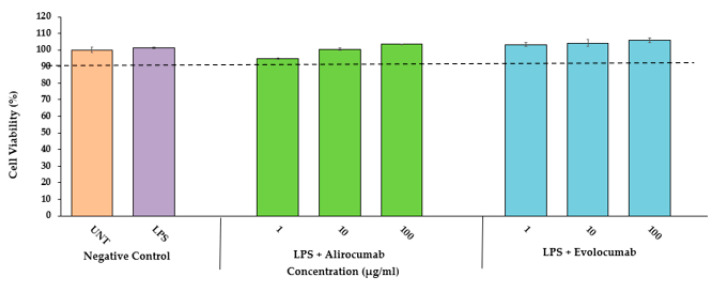
Cytotoxic effects of alirocumab and evolocumab on stimulated HCAEC. Results are presented as percentages (%) of control (untreated cells). Data are expressed as mean ± SD (n = 9). Abbreviation: UNT (Untreated), LPS (Lipopolysaccharides).

**Figure 2 ijms-24-05098-f002:**
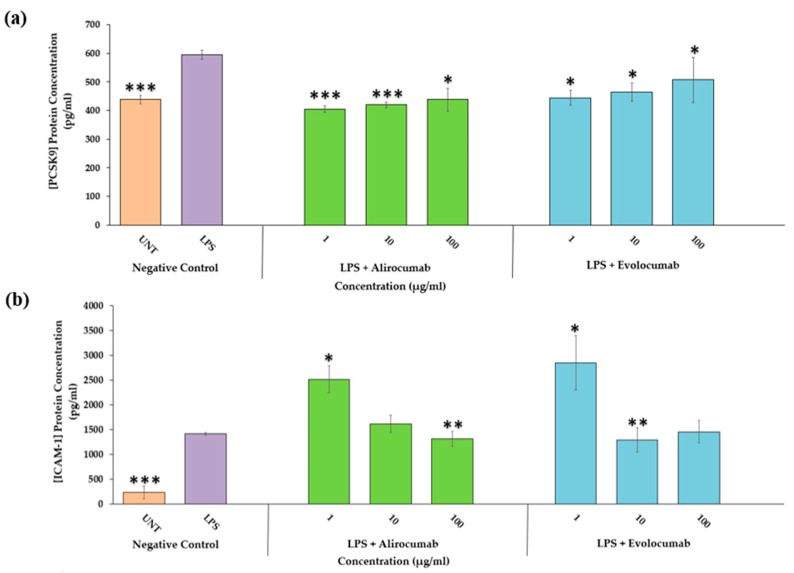
Effects of PCSK9 inhibitors on (**a**) PCSK9, (**b**) ICAM-1, (**c**) E-Selectin, (**d**) IL-6 protein expression in LPS-stimulated HCAEC. Data are expressed as mean ± SD (n = 9). Statistical analysis: ANOVA, post-hoc with Bonferroni correction; * *p* < 0.05, ** *p* < 0.01, and *** *p* < 0.001 compared to HCAEC incubated with LPS alone. Abbreviation: UNT (Untreated), LPS (Lipopolysaccharide).

**Figure 3 ijms-24-05098-f003:**
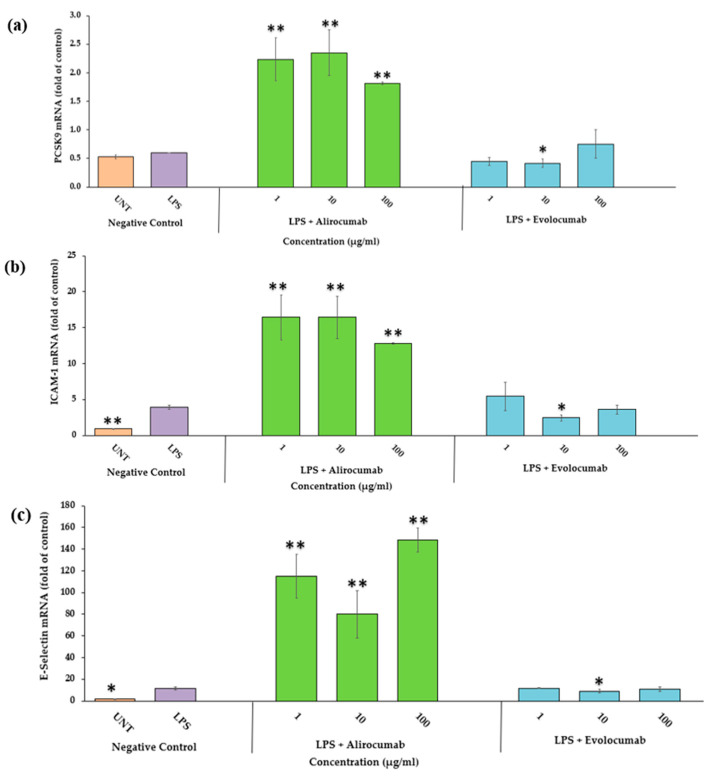
Effects of PCSK9 inhibitors on (**a**) PCSK9, (**b**) ICAM-1, (**c**) E-Selectin, (**d**) IL-6 gene expression in LPS-stimulated HCAEC. Data are expressed as mean ± SD (n = 9). Statistical analysis: ANOVA, post-hoc with Bonferroni correction; * *p* < 0.05, ** *p* < 0.01, and *** *p* < 0.001 compared to HCAEC incubated with LPS alone. Abbreviation: UNT (Untreated), LPS (Lipopolysaccharide).

**Figure 4 ijms-24-05098-f004:**
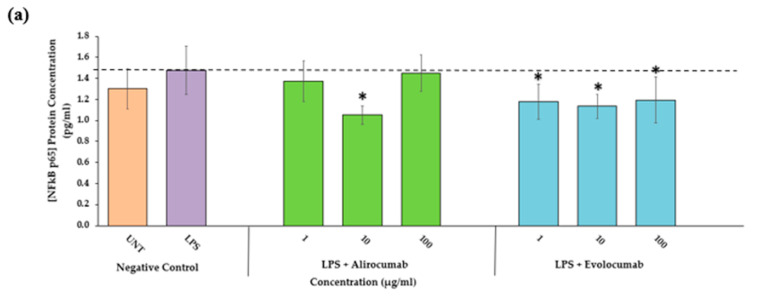
Effects of PCSK9 inhibitors on (**a**) NF-ĸB p65 in nuclear lysate protein expression and (**b**) NF-ĸB p65 gene expression in cell pellets in LPS-stimulated HCAEC. Data are expressed as mean ± SD (n = 9). Statistical analysis: ANOVA, post-hoc with Bonferroni correction; * *p* < 0.05 compared to HCAEC incubated with LPS alone. Abbreviation: UNT (Untreated), LPS (Lipopolysaccharide).

**Figure 5 ijms-24-05098-f005:**
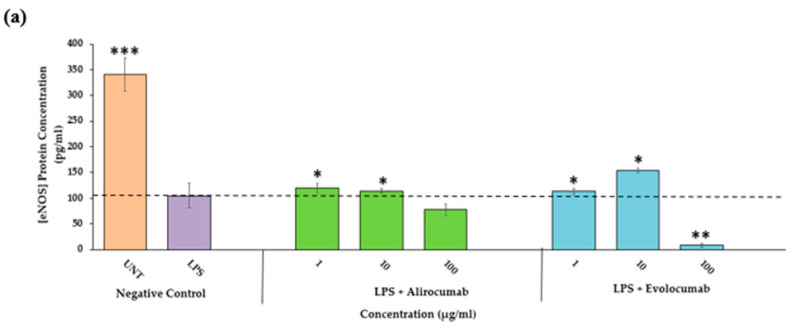
Effects of PCSK9 inhibitors on (**a**) eNOS in cell lysate protein expression (**b**) eNOS gene expression in cell pellets in LPS-stimulated HCAEC. Data are expressed as mean ± SD (n = 3). Statistical analysis: ANOVA, post-hoc with Bonferroni correction; * *p* < 0.05 ** *p* < 0.01 and *** *p* < 0.001 compared to HCAEC incubated with LPS alone. Abbreviation: UNT (Untreated), LPS (Lipopolysaccharide).

**Figure 6 ijms-24-05098-f006:**
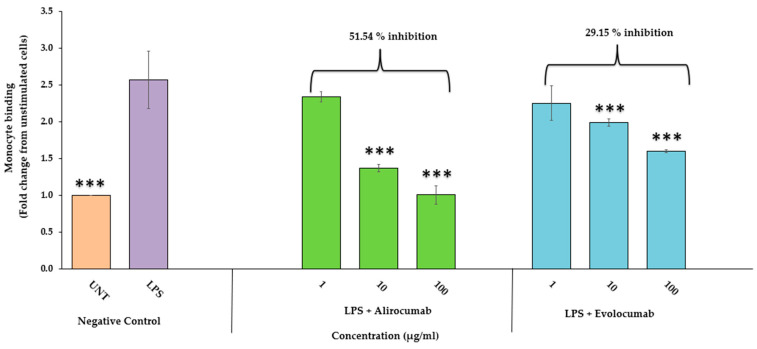
Effects of PCSK9 inhibitors (1–100 µg/mL) on monocyte-endothelial cell binding assay by LPS-stimulated HCAEC. Data are expressed as mean ± SD (n = 9). Statistical analysis: ANOVA, post-hoc with Bonferroni correction; *** *p* < 0.001 compared to HCAEC incubated with LPS alone. Abbreviation: UNT (Untreated), LPS (Lipopolysaccharide).

**Table 1 ijms-24-05098-t001:** Percentage of inhibition of PCSK9, inflammation, endothelial activation, NF-ĸB p65, and eNOS biomarkers by PCSK9 inhibitors based on area under the curve (AUC) analysis.

	PCSK9 %		ICAM-1 %		E-Selectin %		IL-6 %		NF-κB p65 %		eNOS %	
	Inhibition		Inhibition		Inhibition		Inhibition		Inhibition		Increment	
	P	G	P	G	P	G	P	G	P	G	P	G
Evolocumab	5.7	−7.5	−3.3	57.4	3.7	126.9	−132.7	−156.7	21.6	−3.8	65.43	−18.6
Alirocumab	14.0	−129.4	−5.4	−791.5	11.9	−6761.7	−119.2	−1373.1	16.0	−103.1	16.24	−63.5

Abbreviation: P (Protein), G (Gene), Proprotein convertase subtilisin-Kexin type 9 (PCSK9), Intercellular adhesion molecule 1 (ICAM-1), Interleukin-6 (IL-6), Nuclear factor kappa B (NF-ĸB) p65, Endothelial nitric oxide synthase (eNOS).

## Data Availability

Data are contained within the article.

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
