# Peer review of "PCSK9 Inhibitors Reduce PCSK9 and Early Atherogenic Biomarkers in Stimulated Human Coronary Artery Endothelial Cells"

_ijms, 2023, doi:10.3390/ijms24065098_

Round 1
Reviewer 1 Report
The authors investigated the role of PCSK9 inhibitors in endothelial cells.
All experiments are logical and well-designed.
The results were thoroughly discussed.
There is a minor comment:
line 344-345 You wrote that monocyte adhesion is an initial stage of atherosclerosis. However, it is a key event leading to vascular inflammation. Moreover, this event may not be involved in atherosclerosis.
Reviewer 2 Report
Congratulations to the Authors for this interesting paper.
I have just some important questions: ¿What is the mechanism that explain your results? ¿How monoclonal antibodies vs PCSK9 without access to the cell could suppress transcription or translation of inflammatory molecules? ¿Could you discuss about those points: is an extracellular or intracellular effect?
Reviewer 3 Report
The authors aimed to investigate the effects of PCSK9 inhibitors on PCSK9, targeted early atherogenesis biomarkers, and monocyte binding in stimulated human coronary artery endothelial cells (HCAEC). HCAEC were stimulated with LPS and incubated with Evolocumab or Alirocumab. Protein and gene expression of PCSK9, IL-6, E-selectin, ICAM-1, NF-ĸB p65, and eNOS were measured. The binding of U937 monocytes to endothelial cells capacity was measured by the Rose Bengal method. They found that the anti-atherogenic effects of Evolocumab and Alirocumab are contributed by the downregulation of PCSK9, early atherogenesis biomarkers, and significant inhibition of monocyte adhesion to the endothelial cells via the NF-kB and eNOS pathways. Their results suggest the beyond cholesterol-lowering beneficial effects of PCSK9 inhibitors in impeding atherogenesis during the initial phase of atherosclerotic plaque development.
Comments:
1. The first paragraph is too general. This part should be rephrased.
2. Were the concentrations of the Alirocumab and Evolocumab (1,10, and 100 μg/ml) comparable to the in vivo serum concentrations measured in Alirocumab and Evolocumab treated patients?
3. According to the results, there are significant differences in the effect of the two PCSK9 inhibitor monoclonal antibodies on protein and gene expression of PCSK9, IL-6, E-selectin, ICAM-1, NF-ĸB p65, and eNOS. However, based on the previous data, the cause of it is unclear. Since the mechanism of action is the same, technical problems might be responsible for these differences. Repeated measurements on different cell lines are needed to verify these unexpected findings. The “explanations” mentioned in the Discussion (Ln 242-250) are not acceptable.
4. The Discussion is too long and not focused enough.
5. Treatment of HCAEC: the figure is useless (Ln 397)
6. There are severe typos throughout the manuscript.
7. English needs editing.
Round 2
Reviewer 3 Report
Although the manuscript, especially the Discussion part is still quite long, here is a significant improvement. I accept the responses of the authors.